# Destruction of the vascular viral receptor in infectious salmon anaemia provides *in vivo* evidence of homologous attachment interference

**Maria Aamelfot**[1☯¤], **Johanna Hol Fosse**[1☯], **Hildegunn Viljugrein**[1], **Frieda Betty Ploss**[1], **Sylvie L. Benestad**[1], **Alastair McBeath**[2], **Debes Hammershaimb Christiansen**[3], **Kyle Garver**[4], **Knut Falk**[1]*

1 Norwegian Veterinary Institute, Ås, Norway, 2 Marine Laboratory, Aberdeen, Scotland, United Kingdom, 3 Faroese Food and Veterinary Authority, National Reference Laboratory for Fish Diseases, Tórshavn, Faroe Islands, 4 Fisheries and Oceans Canada Pacific Biological Station, Nanaimo, British Columbia, Canada

☯ These authors contributed equally to this work.
¤ Current address: Norwegian Institute of Public Health, Oslo, Norway
* knut.falk@gmail.com

**Data Availability Statement:** All relevant data are within the manuscript and its Supporting Information files.

## Abstract

Viral interference is a process where infection with one virus prevents a subsequent infection with the same or a different virus. This is believed to limit superinfection, promote viral genome stability, and protect the host from overwhelming infection. Mechanisms of viral interference have been extensively studied in plants, but remain poorly understood in vertebrates. We demonstrate that infection with infectious salmon anaemia virus (ISAV) strongly reduces homologous viral attachment to the Atlantic salmon, *Salmo salar* L. vascular surface. A generalised loss of ISAV binding was observed after infection with both high-virulent and low-virulent ISAV isolates, but with different kinetics. The loss of ISAV binding was accompanied by an increased susceptibility to sialidase, suggesting a loss of the vascular 4-*O*-sialyl-acetylation that mediates ISAV attachment and simultaneously protects the sialic acid from cleavage. Moreover, the ISAV binding capacity of cultured cells dramatically declined 3 days after ISAV infection, accompanied by reduced cellular permissiveness to infection with a second antigenically distinct isolate. In contrast, neither infection with infectious haematopoietic necrosis virus nor stimulation with the viral mimetic poly I:C restricted subsequent cellular ISAV attachment, revealing an ISAV-specific mechanism rather than a general cellular antiviral response. Our study demonstrates homologous ISAV attachment interference by de-acetylation of sialic acids on the vascular surface. This is the first time the kinetics of viral receptor destruction have been mapped throughout the full course of an infection, and the first report of homologous attachment interference by the loss of a vascular viral receptor. Little is known about the biological functions of vascular *O*-sialyl-acetylation. Our findings raise the question of whether this vascular surface modulation could be linked to the breakdown of central vascular functions that characterises infectious salmon anaemia.

**Funding:** The work was supported by the Norwegian Research Council (Project numbers 207024 and 244110/HAVBRUK2 and 302191/FRIPRO) and internal funding from the Norwegian Veterinary Institute. The experimental infection of fish was supported by the European Community's Seventh Framework Programme (FP7, 2007-2013) Research Infrastructure Action (Grant FP7-228394). The funders had no role in study design, data collection and analysis, decision to publish, or preparation of the manuscript.

**Competing interests:** The authors have declared that no competing interests exist.

## Author summary

Viral interference, also referred to as superinfection exclusion, is a process that supports viral genome integrity and protects the host from overwhelming infection. Here, we demonstrate that infection of Atlantic salmon with infectious salmon anaemia virus (ISAV) results in the destruction of the viral vascular surface receptor, thus preventing virus attachment. We also observed that the loss of viral receptor strongly restricted the extent of a second ISAV infection in cultured cells, suggesting viral interference. To our knowledge, this is the first time the kinetics of viral receptor destruction has been explored in an infected host. This is important, because we know little of how such responses develop in animals and humans. Our study therefore improves the general understanding of how viral infections progress. Finally, our findings raise the question of whether modulation of the vascular surface by ISAV and other viruses may contribute to the pathogenesis of viral disease.

## Introduction

Infectious salmon anaemia virus (ISAV) is the causal agent of infectious salmon anaemia [1,2]. The disease is a generalised and lethal condition of farmed Atlantic salmon, *Salmo salar* L. with terminal stages characterised by severe anaemia, bleedings, and circulatory disturbances [3]. The exact mechanisms by which the virus causes disease remain incompletely understood. The infection is initiated by an early viral replication phase in the mucosal surface epithelium of the skin and gills, followed by dissemination of the virus to the circulation by a hitherto uncharacterised route [4,5]. Detection of viral proteins by immunostaining has identified that the main target cells of infection are endothelial cells, lining the vasculature of all organs [5]. In addition, circulating erythrocytes sequester active viral particles in infected fish [5,6].

ISAV is an enveloped, segmented, negative-stranded RNA-virus within the genus *Isavirus* of the family *Orthomyxoviridae* [2,7,8]. The ISAV genome consists of eight single stranded negative sense RNA segments that encode at least ten distinct proteins, including four major structural proteins. Two glycoproteins, the haemagglutinin-esterase (HE) and the fusion protein, form spikes that protrude from the viral envelope. The matrix- and nucleo-proteins remain inside the envelope, where the nucleoprotein, the three polymerases and the genomic RNA form the ribonucleoprotein complex [3].

HE is a dual function protein, with both receptor-binding and receptor-destroying activities. The attachment to the cellular surface is mediated by 4-*O*-acetylated sialic acids [5,9] and can be inhibited by serum glycoproteins from rabbits, guinea pigs, and horses, all containing high levels of 4-*O*-acetylated sialic acids [9]. The only other viruses known to use this specific glycan for attachment are found within the species *Murine coronavirus*, of which the mouse hepatitisvirus is the best characterised [10–12]. However, a range of viruses attach to 9-*O*-acetylated sialic acids, including influenza C and D [13,14], as well as many beta-coronaviruses, including SARS CoV-2 [15–17].

The tissue distribution of sialic acid virus receptors can be mapped in tissues using different probes, including lectins that bind specific sialic acids (lectin histochemistry), purified whole virus, and haemagglutinin preparations (virus histochemistry, VHC, illustrated in Fig 1A) [18,19]. Establishing such methodology for ISAV, we previously showed that the distribution of the ISAV cellular receptor mirrored the distribution of infected cells in Atlantic salmon, as demonstrated by the expression of viral antigens by immunohistochemistry or

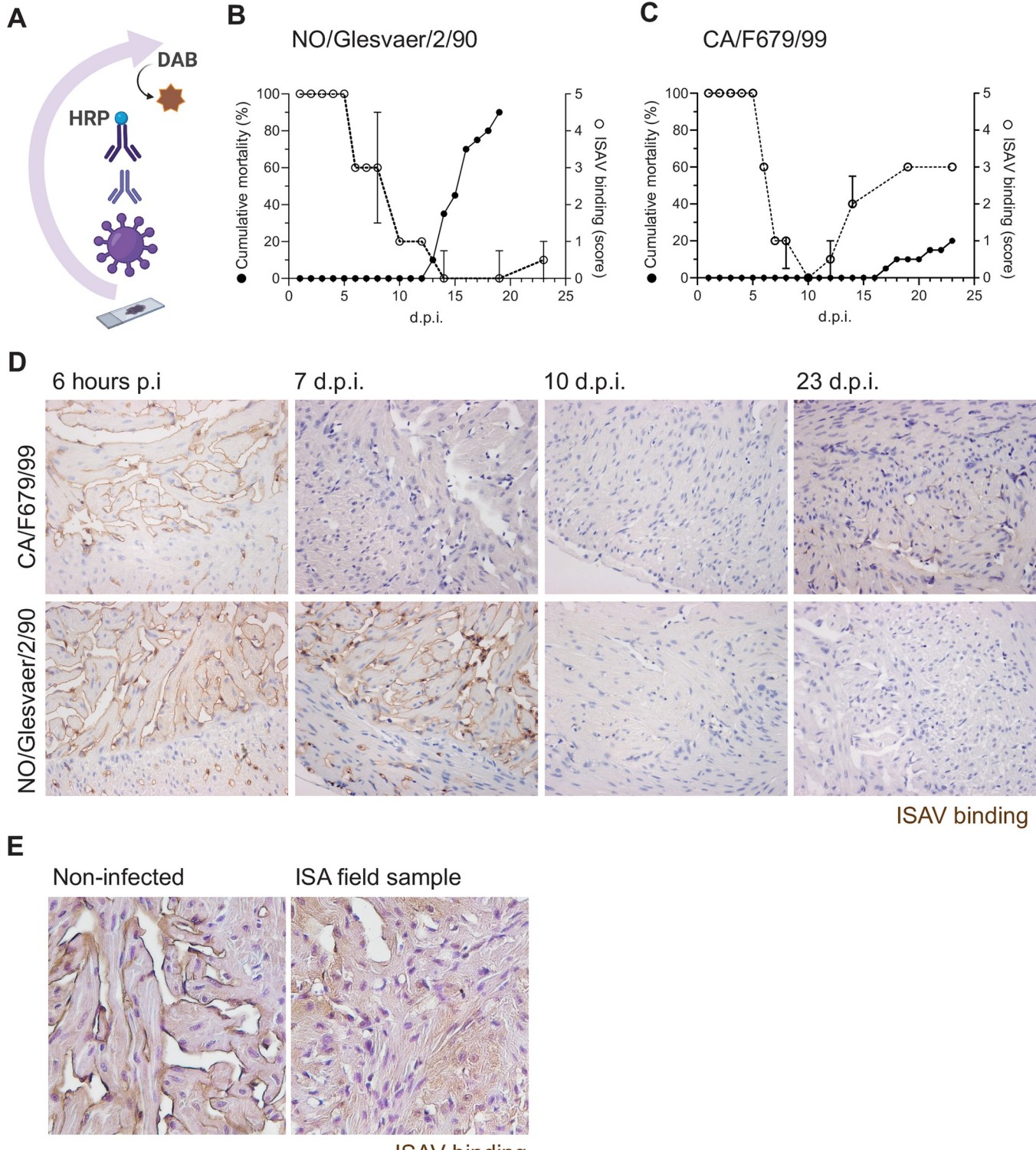

**Fig 1. Vascular endothelial cells in ISAV-infected fish become refractory to further ISAV attachment.** (A) Tissue expression of the ISAV receptor was mapped by virus histochemistry using ISAV antigen as the primary probe. Binding of the probe was detected by an antibody targeting the virus, a secondary HRP-labelled antibody, and DAB. The illustration was created in Biorender.com. (B-C) Cumulative mortality (black dots and lines) and ISAV binding (open circles, stippled line, data points show median scores +/- 95% confidence intervals) after infection with two different ISAV isolates: (B) High-virulent NO/ Glesvaer/2/90 and (C) low-virulent CA/F679/99. Mortality data from this trial have been published before [35], but are included for context. The loss of ISAV binding was calculated by virus histochemistry on tissue sections from fish harvested 1, 2, 3, 4, 5, 6, 7, 8, 10, 12, 14, 19, and 23 d.p.i. (n = 4 fish per group). (D)

Representative micrographs of virus histochemistry in hearts from infected fish at given time points. (E) Representative images of virus histochemistry in hearts from moribund fish collected during Norwegian infectious salmon anaemia outbreaks and non-infected fish (n = 3 fish per group). (D-E) Positive binding is identified by DAB (ISAV binding, brown).

immunofluorescence, i.e. endothelial and epithelial cells [5]. In this context, we also characterised a monoclonal antibody (10E4) that turned out to be a pan-endothelial and erythrocyte marker in Atlantic salmon, and provided evidence that this antibody recognises an epitope of the sialic acid carrying the ISAV receptor that is distinct from the ISAV-targeted 4-*O*-acetyl group [20].

In addition to the receptor binding domain, HE harbours a receptor destroying enzyme (RDE); a sialate-4-*O*-esterase that releases ISAV-receptor interactions [9,21,22]. In solution, both 4- and 9-*O*-acetylated-sialic acids can be hydrolysed by the ISAV esterase; However, when the substrate is attached to an underlying glycan, the ISAV esterase only targets 4-*O*-acetylated sialic acid [9]. So far, the role of the ISAV RDE in infection and pathogenesis has not been characterised. However, RDEs of other sialic acid-binding viruses contribute to mucus penetration, cellular uptake, uncoating, and release of new virus particles from infected cells [23–26]. Interestingly, RDE activity also limits viral superinfection *in vitro* by destroying virus receptors on infected cells [27–30], leading to viral attachment interference.

Viral interference occurs when infection with one virus mediates cellular resistance to infection by a second virus [31]. Interference may be homologous, mediated by the same virus, or heterologous, by another virus [31,32]. Functionally, viral interference is thought to prevent replication of two or more viral genomes in the same cell, thereby limiting reassortment and recombination of viral genomes and contributing to viral genome stability and integrity over time [33]. In the context of disease pathogenesis, viral interference may furthermore limit virus replication and load, thus preventing or delaying excessive damage to the host. Viral interference was originally described for plant viruses in 1929 [34], and similar observations were made for a plethora of animal viruses in the 1940s and 1950s [35]. During the 1970s, 1980s, and 1990s, several *in vitro* studies demonstrated that infected cells became resistant to a second infection. Later, homologous attachment interference in cell culture by destruction of the cellular virus receptor by RDE/neuraminidase was reported for influenza A virus [27], Sendai virus [28], human parainfluenza virus type 3 [29], and Newcastle disease virus [30]. However, despite extensive studies of attachment interference in cell culture systems, only a few studies have provided evidence of this phenomenon in tissues from infected individuals [36,37], and none have studied its kinetics during the course of an infection.

Here, we present evidence that viral receptor destruction is a consistent and wide-spread finding in infectious salmon anaemia. We map the kinetics of receptor loss during experimental infection with two ISAV isolates of different virulence and correlate the loss of the ISAV receptor to the expression of viral antigens in tissues. The loss of the ISAV receptor from the vascular surface was caused by a loss of sialic acid 4-*O*-acetylation, supporting that it could be mediated by viral RDE/esterase hydrolysis. Furthermore, we observed a similar effect in cell culture that mediated resistance to a second infection with ISAV, suggesting viral attachment interference.

# Results

## Experimental infection

The immersion challenge in Atlantic salmon that provided material for this study has been described in detail elsewhere [4,5,38,39]. Briefly, after infection with the high-virulent NO/

Glesvaer/2/90 isolate [1], mortality started 13 days post infection (d.p.i.) and progressed to 100% by 21 d.p.i. in the control tank used for monitoring mortality (Fig 1B). A few surviving fish in the sampling tank allowed sampling at day 23. In contrast, infection with the low-virulent CA/F679/99 isolate resulted in a much less severe course of infection, suggesting lower virulence [38,39]. In the group of fish infected with CA/F679/99, mortality ensued 17 d.p.i. and increased to 20% at the end of the trial, 23 d.p.i. (Fig 1C).

## Vascular endothelial cells in ISAV-infected fish become refractory to further ISAV attachment

In the first step of the infectious cycle, ISAV binds 4-*O*-acetyl-sialic acid on the host cell surface [5,9]. The ability of cells to support such attachment can be mapped in formalin-fixed paraffin-embedded tissue sections by virus histochemistry (Fig 1A) [5]. We observed that the characteristic ISAV attachment to endothelial cells that is observed in non-infected fish [5] was retained at early time points after ISAV infection, but was lost or severely reduced as the infection progressed (Figs 1B–1D and S1). The loss of virus binding followed different kinetics in fish infected with NO/Glesvaer/2/90 and CA/F679/99 (Figs 1B–1D and S1). Virus binding was strongly reduced in samples collected 7 d.p.i. with CA/F679/99, but a weak signal started to reappear 14 d.p.i. and was clearly visible 23 d.p.i. (Fig 1C and 1D). In contrast, loss of binding only became evident 10 d.p.i. with NO/Glesvaer/2/90 and was not restored (Fig 1B and 1D). In both groups, the loss of virus binding capacity preceded the onset of mortality (Fig 1B and 1C). The loss of ISAV binding was uniform and global, with the same pattern observed in heart (Fig 1D), kidney (S1A Fig), and liver (S1B Fig). Further elaborating on our observations in experimentally infected fish, we performed virus histochemistry on diagnostic heart samples from three Norwegian infectious salmon anaemia outbreaks. We observed a strong reduction in virus binding (Fig 1E), although with a slightly patchier appearance than in experimentally infected fish, where the loss was complete. Together, our observations suggest a global modulation of the vascular surface in infectious salmon anaemia that limits further ISAV attachment.

## Vascular sialic acids are preserved in infected fish, but loose the 4-*O*-acetylation that constitutes the ISAV receptor

Confirming the involvement of the viral receptor, we used recombinant ISAV HE as a probe and observed that it bound sections in non-infected, but not infected fish (Fig 2A). We have previously generated a monoclonal antibody (10E4) that reacts with Atlantic salmon endothelial cells from all branches of the circulation by binding a non-acetylated epitope closely associated with the ISAV 4-*O*-acetyl-sialic acid receptor [20]. This is most likely an epitope of the sialic acid itself, as the 10E4 staining in tissues from healthy fish was abolished by sialidase, but only when the protective 4-*O*-acetylation was removed prior to sialidase treatment by mild alkaline hydrolysis (saponification) [20,40,41]. When tissue sections from experimentally infected fish were immunostained with 10E4, no reduction in signal was observed, compared to controls (Figs 2B and S2A), despite the loss of virus binding capacity. This suggests persistence of a sialic acid that can no longer bind ISAV. We next explored if the lack of ISAV binding could be explained by de-acetylation of the sialic acid, by applying the principle described above and illustrated in Fig 2C. When tissue sections were incubated with bacterial sialidase prior to staining, we observed a marked reduction in 10E4 signal in vascular beds of heart and kidney samples from infected fish that had lost the capacity to bind ISAV (Figs 2D–2E, S2B and S2C). In contrast, tissues from non-infected fish that retained the capacity to bind ISAV, required prior saponification to render the vascular 10E4 epitope susceptible to sialidase (Figs 2D–2E, S2B and S2C). This shows that the loss of ISAV attachment in infected fish is

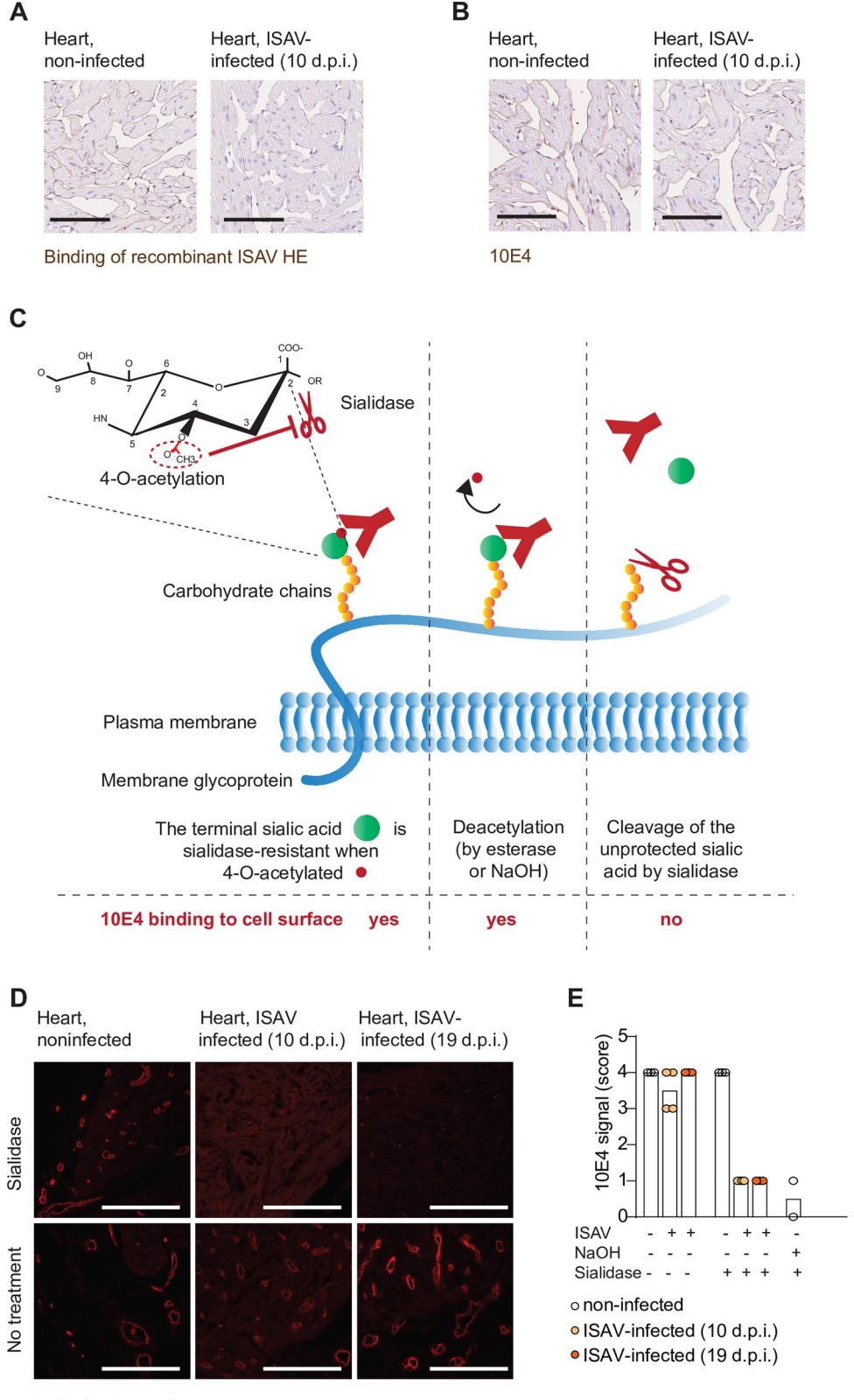

**Fig 2. Vascular sialic acids are preserved in infected fish, but lose the 4-O-acetylation that constitutes the ISAV receptor.** (A) Binding of recombinant ISAV HE (brown) in heart of non-infected and Glesvaer/2/90-infected fish. (B)

Immunostaining for the sialic acid backbone of the ISAV receptor (10E4, brown) in heart of non-infected and Glesvaer/2/90-infected fish. (C) Illustration of the principle used to investigate 4-*O*-acetylation. The 4-*O*-acetylation (red dot) protects the sialic acid (green dot) against cleavage (left panel). When the modification is removed by either NaOH or an esterase, the 10E4 epitope is preserved (middle panel), but becomes susceptible to sialidase cleavage (right panel). (D) Representative micrographs of immunofluorescent staining by 10E4 in hearts of experimentally infected and non-infected fish (n = 4 fish per group), illustrating that ISAV infection renders the 10E4 epitope susceptible to sialidase. Scale bars are 100 μM. (E) Scoring of immunostainings showed in (D). Dots show scores in individual fish, and bars show median score in group.

associated with a gain of sialic acid susceptibility to sialidase, suggesting that the observed loss of ISAV attachment is due to a global vascular loss of its receptor, the 4-*O*-acetylation that also protects sialic acids from sialidase cleavage.

## ISAV infection of cultured cells causes loss of the ISAV receptor and induces attachment interference

To test if the observed loss of ISAV attachment prevented infection or could be compensated for by other attachment strategies, we serially infected Atlantic salmon kidney (ASK [42]) cells with antigenically distinct ISAV isolates that could be distinguished by genotype-specific HE-reactive antibodies (S3 Fig). Similar to findings in tissues from experimentally infected fish, ASK cell infection with the ISAV isolate CA/NBISA05/98 restricted the subsequent attachment of NO/Glesvaer/2/90 (Fig 3A–3C) to a level that resembled the effect of removing 4-*O*-acetylation by saponification (Fig 3C). The loss of ISAV attachment was accompanied by a dose-dependent interference with a second ISAV infection (Fig 3D). Similar levels of interference were observed when the order of the virus isolates was reversed (Fig 3E). Together, these results suggest that the loss of ISAV attachment that accompanies a primary infection renders the cells refractory to superinfection and therefore represents a viral interference phenomenon.

## The loss of the cellular ISAV receptor is associated with the production of viral antigens and levels of circulating virus

The esterase portion of ISAV HE is a viral RDE that deacetylates 4-*O*-acetylated sialic acids [9,21,22]. We observed that the loss of ISAV attachment in hearts of infected fish coincided with extensive production of viral antigens by endothelial cells, as demonstrated by immuno-histochemical detection of ISAV nucleoprotein (Fig 4A and 4B). Likewise, immunofluorescent staining of blood smears from infected fish revealed substantial coating of red blood cells with ISAV HE that followed a similar time course (Fig 4A and 4B). Comparing virus histochemistry, immunohistochemistry, and blood smear immunofluorescence scores in individual fish infected with NO/Glesvaer/2/90 (Fig 4C) or CA/F679/99 (Fig 4D) by Spearman's correlation and bootstrapping, we observed a negative relationship between immunohistochemistry scores and virus histochemistry scores (rho = -0.80; 95% confidence interval (95% CI): -0.89 to -0.73) and between blood smear immunofluorescence scores and virus histochemistry scores (rho = -0.75; 95% CI: -0.87 to -0.66). To reduce the impact of overall time trends, we also compared differences between scores and the mean score from the respective previous time point. Using this method, a decrease in virus histochemistry score from the mean score of previous time was associated with an increase in immunohistochemistry score from the mean score one time step earlier (rho = -0.45; 95% CI: -0.67 to -0.27), as well as an increase in blood smear immunofluorescence score from the mean score two (but not one) time steps earlier (rho = -0.33; 95% CI: -0.53 to -0.14). Similarly, in ASK cells, the loss of virus attachment followed the temporal expression of ISAV HE (Fig 4E).

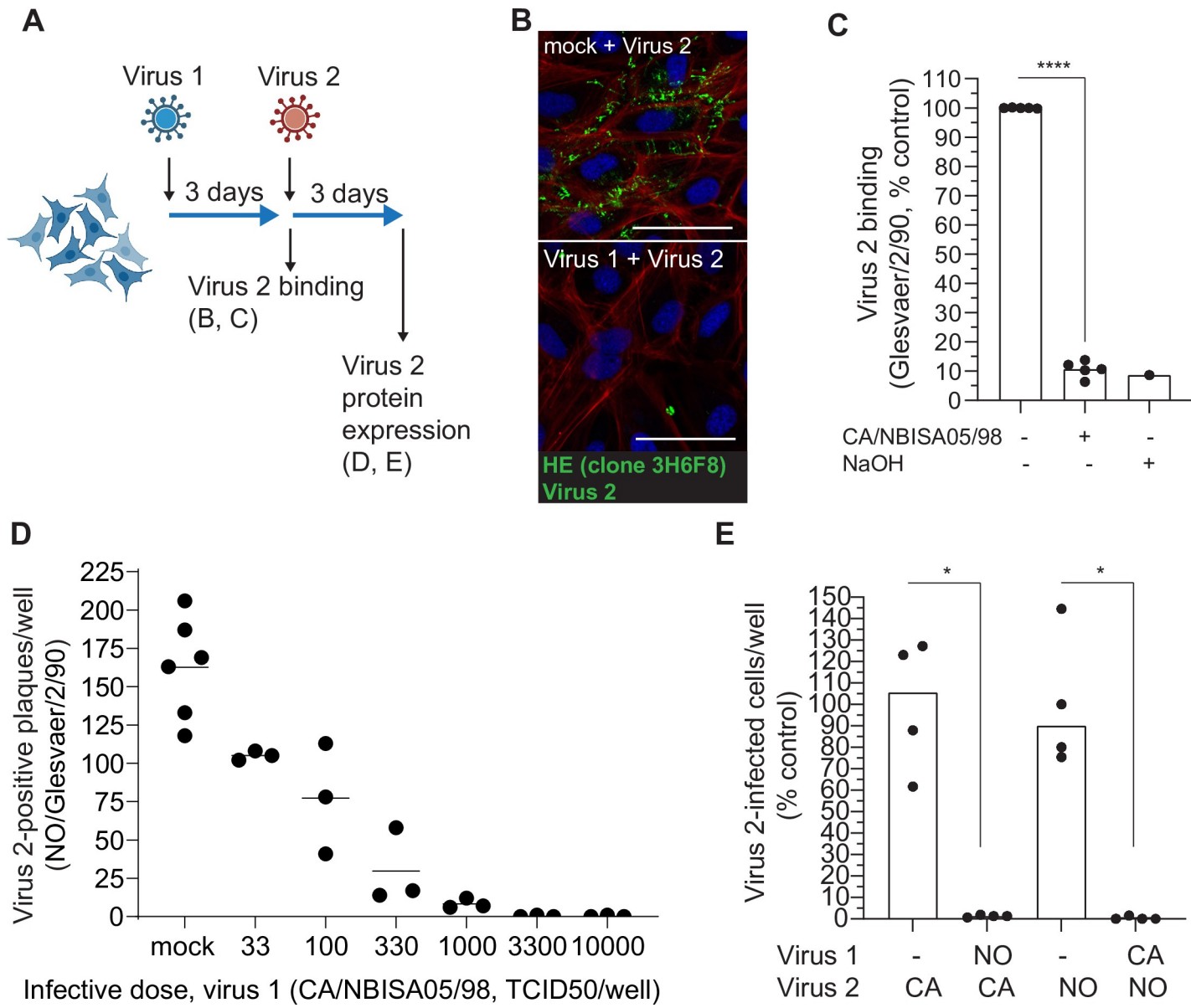

**Fig 3. ISAV infection of ASK cells induces attachment interference.** (A) ASK cells were infected with two antigenically distinct ISAV isolates 3 days apart, and binding and protein expression of the second isolate (virus 2) was evaluated by strain-specific antibodies. The illustration was created in Biorender.com. (B) NO/Glesvaer/2/90 was allowed to bind the surface of non-infected (mock + virus 2) and CA/NBISA05/98-infected (virus 1 + virus 2, 3 d.p.i.) ASK cells for one hour, before extensive washing and detection by immunostaining for ISAV HE (green). Phalloidin (red) and Hoechst 33342 (blue) were used to visualise actin fibres and nuclei. Scale bars are 50 μM. Micrographs are representative of two experiments. (C) Virus binding assay quantifying NO/Glesvaer/2/90-binding to non-infected and CA/NBISA05/98-infected (3 d.p.i.) ASK cells. Cells treated with NaOH to remove the ISAV receptor are included as negative controls. Dots show normalised means from four independent experiments. Bars show the mean of means. ****$p<0.0001$, paired two-tailed t test. (D-E) ISAV infection interferes with a second ISAV infection 3 days later. (D) Counts of NO/Glesvaer/2/90 (virus 2) HE-positive plaques 3 d.p.i. in ASK cells infected with serial dilutions of CA/NBISA05/98 (virus 1) 3 days prior to the second infection. Dots show values from individual wells in one experiment. Bars show mean values. (E) The effect of infection with virus 1 on virus 2 protein expression 3 days later. (NO/Glesvaer/2/90 [NO], CA/NBISA05/98 [CA]). The numbers of HE-positive plaques per well were counted manually in 10 microscope fields per well, and normalised to the mean of wells not previously infected. Dots show normalised mean values per well in one representative experiment of three. *$p<0.05$, Mann Whitney U.

These results show that the loss of the cellular ISAV receptor correlates closely with the cellular production of viral proteins, suggesting a possible mechanism by which de-acetylation could take place.

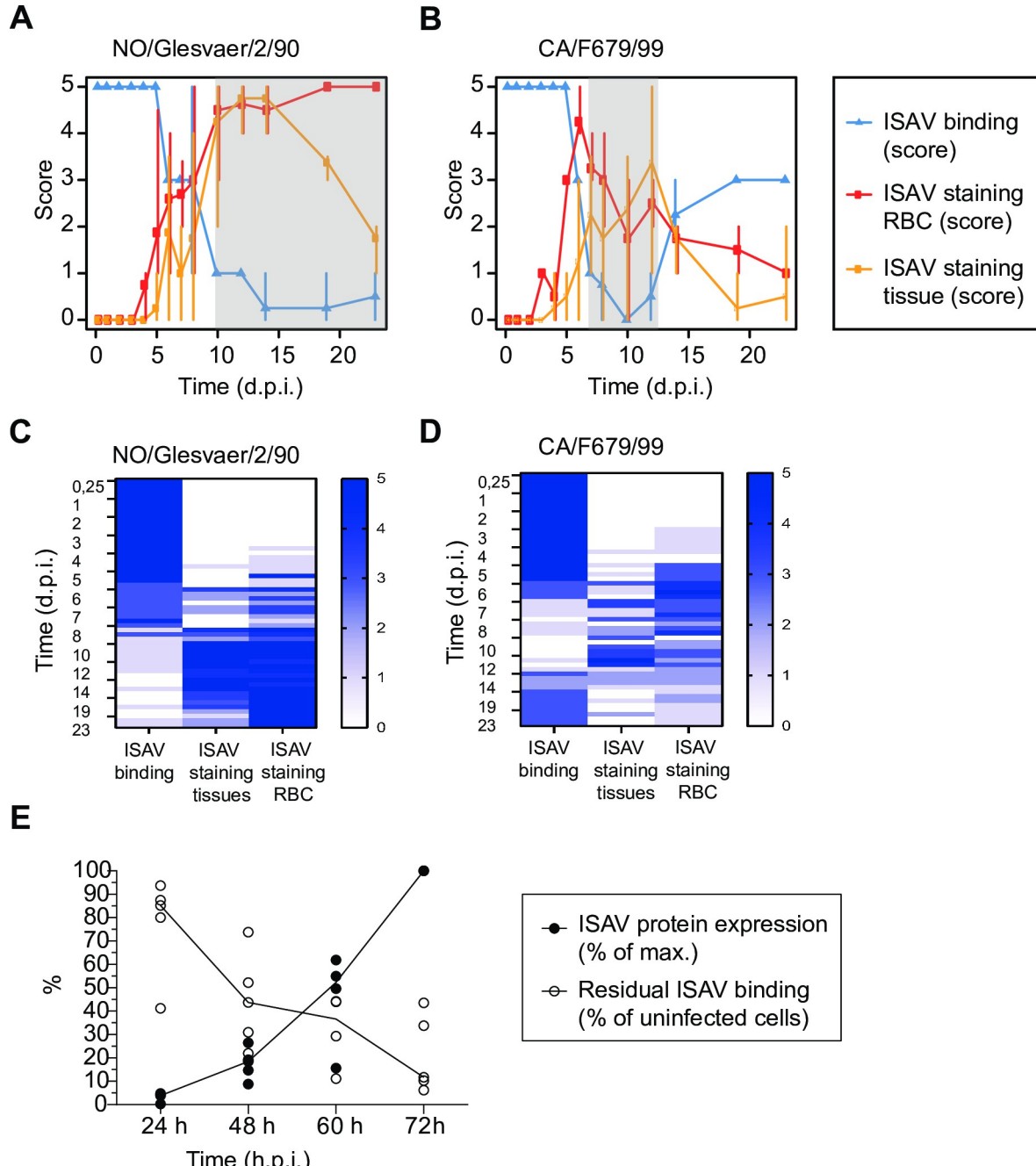

**Fig 4. The loss of the cellular ISAV receptor correlates to circulating virus levels and cellular production of viral proteins.** (A-D) Semi-quantitative scoring of virus histochemistry (ISAV binding, blue), immunohistochemical detection of ISAV NP in heart sections (ISAV staining tissue, yellow), and immunofluorescent staining of ISAV HE in blood smears (ISAV staining RBC, red) in fish infected with NO/Glesvaer/2/90 and CA/F679/99. Time points with individual fish VHC scores = /<1 are indicated in grey. Graphs in (A-B) show mean and range of scores and heat maps in (C-D) show stacked scores in individual fish. (E) Cellular ISAV HE production measured by a cell-based ELISA (black dots) and inhibition of ISAV antigen binding to infected fixed cells by virus binding assay (open circles). Both assays used strain-specific antibodies to ISAV HE. Data points show normalised mean values of five experiments in three independent runs: Three experiments infected cells with CA/NBISA05/98 and measured NO/Glesvaer/2/90 binding. Two experiments infected cells with NO/Glesvaer/2/90 and measured CA/NBISA05/98 binding. Lines connect median values.

## The general host response to viral infection does not restrict ISAV attachment

Reduced viral binding has also been observed in H5N1 highly pathogenic avian influenza. The reduced H5N1-binding coincided with the cellular production of the antiviral factor Mx, and the authors suggested that it could be mediated by the intrinsic cellular antiviral response [36]. Not surprisingly, the expression of Mx and type I interferons was strongly stimulated in infected fish in the current study and correlated to viral RNA levels (reported in [38]). To test if the reduction in ISAV binding could be caused by a general host response to the viral infection, we performed virus histochemistry on tissues from Atlantic salmon experimentally infected with another endotheliotropic RNA virus, infectious haematopoietic necrosis virus (IHNV). We found that IHNV infection did not restrict vascular ISAV attachment (Fig 5A). Moreover, we observed no reduction in ISAV binding to ASK cells after infection with IHNV

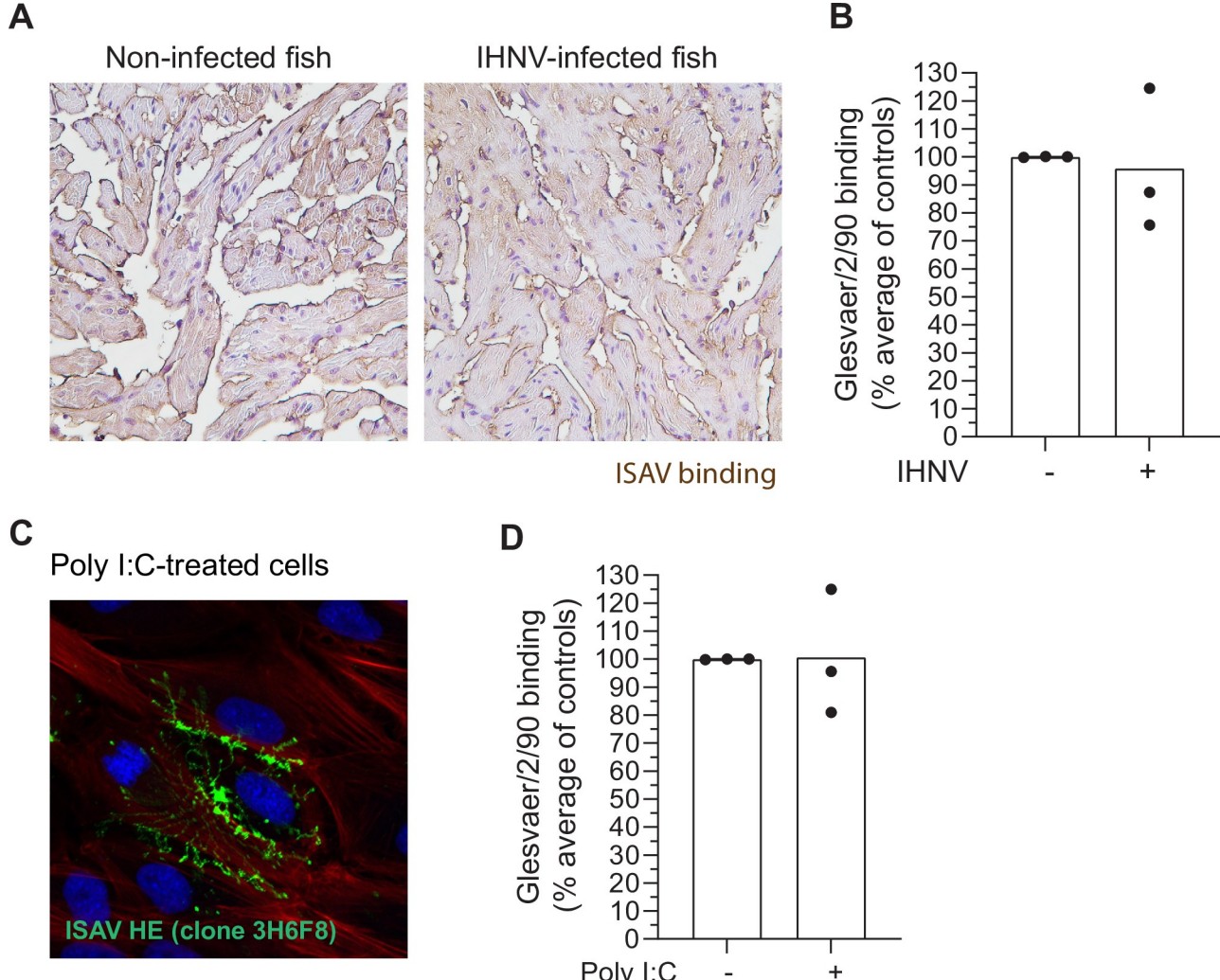

**Fig 5. ISAV binding is not affected by the general antiviral response.** (A) Virus histochemistry in non-infected and IHNV-infected fish (n = 3 fish per group). (B) Virus binding assay in IHNV-infected ASK cells. (C) Representative micrographs visualising ISAV binding in poly I:C-treated ASK cells. (D) Virus binding assay in poly I:C-treated ASK cells. (B and D) Dots show normalised means from three independent experiments. Bars show mean values.

(Fig 5B) or exposure to the viral RNA mimetic Poly I:C (Fig 5C–5D), a known stimulator of antiviral signalling [43].

These findings show that the ISAV receptor is not lost upon a general stimulation of the cellular antiviral response. Rather, the vascular loss of ISAV receptor is specific to ISAV infection, suggesting that it is due to homologous viral interference rather than a consequence of host innate antiviral responses.

## Discussion

In recent years, several studies have reported the *in situ* localisation of specific virus receptors by virus or lectin histochemistry in tissue sections. Here, we examined the ISAV receptor distribution in experimentally ISAV-infected Atlantic salmon by virus histochemistry. Our findings demonstrate that the ISAV cellular receptor disappears from the vascular surface of ISAV-infected Atlantic salmon during the course of infection. This is, to our knowledge, the first time the kinetics of viral receptor disappearance has been mapped *in vivo* in a vertebrate host. We also provide evidence suggesting that this loss is mediated by the viral RDE, causing deacetylation of the 4-*O*-acetylated sialic acid cellular ISAV receptor. Modelling the response in ASK cell cultures suggested that the loss of viral attachment was associated with loss of permissiveness to a second infection, thereby revealing viral interference.

### Loss of the vascular ISAV receptor

Virus histochemistry allows spatial visualisation of viral attachment in morphologically intact tissues, and has been a valuable tool for understanding influenza virus host tropism [18,19,44,45] and ISAV cell and tissue tropism [5]. The method was also applied in a study of H5N1 highly pathogenic avian influenza, reporting a decrease in sialic acids in *ex vivo* infected human and macaque lung biopsies and lungs of infected ferrets and cats, using lectin histochemistry to confirm the loss of sialic acids [36]. The authors hypothesised that the decrease in virus receptors was most likely part of a general antiviral response, but also mentioned the possibility that RDE could be involved. Another study, relying on lectin histochemistry alone, reported a similar loss of sialic acid expression in pigs after infection with three different influenza A variants (H1N1, H1N2, H4N6) [37]. Both these studies reported the loss of the influenza virus receptor at a single time point of infection. Our findings complement these observations by demonstrating *in vivo* viral receptor destruction by ISAV, another member of the *Orthomyxoviridae* family. Our results also extend on previous observations; by mapping the kinetics of the response to permit correlation to viral loads, and by demonstrating that receptor destruction appears to be a relatively early event in infectious salmon anaemia that precedes mortality and morphologically detectable tissue damage.

4-*O*-acetylation of sialic acids protects against sialidase-mediated cleavage [41]. Taking this into account, we first demonstrated that even though the capacity to bind ISAV was lost, we could still detect the sialic acid backbone of the receptor. We then treated the virus histochemistry-negative sections with bacterial sialidase and found that the sialic acid backbone had become susceptible to cleavage in infected fish. Thus, the loss of ISAV binding appears to be caused by specific loss of the 4-*O*-acetyl-group on the ISAV cellular receptor.

Cellular expression of viral RDE has been linked to viral attachment interference in cells infected with several viruses [27–30]. We observed a correlation between the loss of viral attachment and viral antigen expression in tissue and blood. Similarly, the loss of viral attachment to ISAV-infected ASK cultures mirrored the cellular expression of ISAV HE. To this end, we propose that the most likely cause of receptor destruction in ISAV infection is de-acetylation by the ISAV esterase, acting either *in-cis* (on cell surfaces) or *in-trans* (viral particles in

blood). However, our findings do not completely exclude that endogenous sialic acid *O*-acetyl-esterases may be involved in the observed loss of vascular ISAV binding in infected fish. Indeed, this may be challenging to accomplish, given the limited knowledge of this class of molecules, even in mammals [46]. Functional studies based on recombinant expression of ISAV HE may therefore be a more productive approach to firmly establish the role of the viral esterase in the infection-induced loss of the ISAV cellular receptor.

In support of the hypothesised role of the viral esterase, we observed no loss of ISAV attachment in tissues from IHNV-infected fish or IHNV-infected ASK cells. Furthermore, administration of the viral mimetic poly I:C did not alter ISAV attachment to ASK cells. These findings show that the general cellular response to viral infection is not sufficient to cause loss of 4-*O*-acetylation and viral attachment. Our findings in cell culture also argues that the response represents a homologous viral interference phenomenon. Viral interference may be advantageous to maintain viral genome stability by preventing recombination and reassortment between two viruses that infect the same cell. It may also serve to limit the extent of viral infection and thereby the damage to the host.

## Potential implications for biological function and pathology

The generalised loss of the vascular ISAV receptor could in itself be relevant to disease. While some endotheliotropic viruses severely disrupt endothelial cells, others cause little or no morphological disruption and show a peculiar absence of vessel-associated inflammation, despite extensive endothelial cell production of viral antigens and particles (discussed in [47]). Although a single report showed marked ultrastructural damage to hepatic endothelial cells in ISAV-infected Atlantic salmon [48], our experience is that the characteristic functional vascular compromise in severely affected individuals, with bleeding, oedema, and circulatory disturbances [3], may occur despite very little evidence of vascular morphological damage and inflammation [3,5]. Considering that destruction of glycocalyx components contributes to tissue-specific vascular dysfunction in flavivirus infection [49,50], it is reasonable to ask if the observed loss of 4-*O*-sialyl-acetylation affects the vascular function in affected fish.

Sialic acids are abundantly present in secretions and the cellular glycocalyx, in particular on mucosal and vascular surfaces. While a number of microbes exploit these molecules as cellular receptors [46,51–53], the primary biological function of sialic acids is the regulation of cellular interactions that can be broadly divided into two groups: First, sialic acids act as biological masks, shielding recognition sites and limiting cellular interactions. Vasculoprotective examples include the protection against proteolytic damage [54], prevention of aberrant platelet adhesion and activation [55], and promotion of the vascular barrier [56,57]. It is worth noting that the loss of steric hindrance leaves the de-*O*-acetylated sialic acid layer susceptible to cleavage [41,58]. Hence, our findings suggest that the vascular surface in infected fish becomes more susceptible to endogeneous sialidases, which are reported to be upregulated in endothelial cells in response to viral components [43]. While immunostaining did not reveal any change in sialic acid levels in ISAV-infected Atlantic salmon, we cannot exclude that subtle changes could have escaped our detection. Nevertheless, de-acetylation appears to be the predominant change in infected fish.

Second, sialic acids represent biological interaction sites for hormones, lectins, antibodies, and inorganic cations that affect immune recognition. *O*-acetylation may affect such interactions, for example by limiting sialic acid recognition by sialic acid-binding immunoglobulin-like lectins that regulate immune tolerance and cell homing [46,59–61]. Finally, several molecules involved in vascular regulation also undergo sialylations that affect their biological function [62,63]. Thus, modulation of the vascular sialic acid layer has potential to affect a range of cellular functions and contribute to pathology.

Considering the characteristic anaemia in ISAV infected fish, it is also worth noting that enzymatic removal of erythrocyte and platelet sialic acids leads to their rapid disappearance from the blood stream [64–66]. Moreover, in mouse erythroleukaemia cells, the levels of sialyl-9-*O*-acetylation regulate interactions with complement factor H and sialoadhesin, thus affecting complement-activated lysis and sequestration in liver and spleen [59]. Future studies should therefore address if 4-*O*-acetyl-sialic acids on erythrocytes undergo similar changes in response to ISAV infection to those observed in endothelial cells in the current study.

## Concluding remarks

In conclusion, our study reveals homologous attachment interference by the *Orthomyxoviridae* family member ISAV and raises the question of whether the pan-vascular loss of 4-*O*-sialyl-acetylation has biological consequences related to the characteristic pathology of infectious salmon anaemia.

## Methods

### Ethics Statement

The study protocol for experimental infection of fish and its implementation was approved prior to the studies by the Norwegian Food Safety Authority (FOTS ID 4192) or the Pacific Regional Animal Care Committee (AUP# 17–013). The facilities were operated in compliance with Organisation for Economic Co-operation and Development (OECD) principles of Good Laboratory Practice and Guidelines to Good Manufacturing Practice issued by the European Commission or in accordance with the recommendations in the Canadian Council on Animal Care (CCAC) Guide to the Care and Use of Experimental Animals, respectively.

### Cell culture

ASK cells [42] were used between passage 50 and 73. Cell cultures were maintained in L-15 medium (Lonza Bioscience) supplemented with foetal bovine serum (FBS, Lonza Bioscience, 10%), L-glutamine (Lonza Bioscience, 4mM), and penicillin/ streptomycin/ amphotericin (Lonza Bioscience, 1%) or gentamicin (Lonza Bioscience, 50 μg/mL). The cells were cultured at 20˚C and were split 1:2 every 14 days. For experiments, cells were seeded at a density of approximately 10,000 cells/cm$^2$ in 96-well cell culture plates and allowed to reach confluence. For imaging, cells were seeded at 50,000 cells/cm$^2$ in polymer-coated 8-well μ-slides (Ibidi GmbH). After viral infection, cells were maintained at 14–15˚C. Culture conditions for CHSE-214 cells [67] were as described for ASK cells, except using 5% FBS and splitting 1:3 when confluent.

### Virus and antigen preparations

The high virulent NO/Glesvaer/2/90 and low virulent CA/F679/99 isolates were used for the experimental infections. For cell culture experiments, a Canadian isolate CA/NBISA05/98 was also used, because it is antigenically distinct from NO/Glesvaer/2/90, enabling separate detection in antibody-based assays. Infectious haematopoietic necrosis virus (IHNV) infection was by the Canadian isolate BC93-057 [68]. ISAV virus stocks were propagated in ASK cells. Virus titres were determined by infection of ASK cells, immunofluorescent staining of ISAV HE or NP protein or IHNV protein N, and calculation of the 50% tissue culture infective dose (TCID$_{50}$) by the Spearman-Kärber method as previously described [2]. Virus stocks were propagated in 75–225 cm$^2$ tissue culture flasks. Supernatants were harvested when cytopathic effects were obvious, 7–28 d.p.i. Antigens for virus histochemistry and virus binding assays

were prepared by collecting membrane fractions of infected ASK cells as previously described [5]. Hemagglutination titres were determined by incubating serial dilutions of antigen preparations with 0.75–1% equine erythrocyte suspensions in 96-well V-bottom microtitre plates as previously described [2].

## Generation of recombinant ISAV HE

Codon optimised sequences of the open reading frame encoding ISAV HE, corresponding to isolate NO/Finnmark/NVI-70-1250/2020 (Genbank accession UGL76651.1) was synthesised and inserted in the pcDNA3.1 (+) vector commercially, delivered transfection-ready (Gene-Cust, Boynes, France). Monolayers of CHSE-214 cells were cultured until 90–100% confluent and detached by Trypsin EDTA (Lonza Bioscience). 4 transfection reactions, each with $10^6$ cells and 10 μg plasmid DNA, were performed, using the Neon 100 μL Transfection System (Invitrogen, Waltham, MA, USA, three pulses at 1600 V and 10 ms width). Non-transfected cells were used as controls. The transfected cells were pooled and incubated 24 hours in antibiotic free medium, then another 24 hours in culture medium. Membrane fractions were collected as previously described [5].

## Fish and experimental infection

The analysed material was derived from a previously described infection trial [38,39]. Briefly, 291 unvaccinated Atlantic salmon presmolt (mean weight 133g) were infected by immersion in NO/Glesvaer/2/90 or CA/F679/99 ($10^4$ TCID$_{50}$/mL) (or the corresponding dilution of uninfected tissue culture medium) for two-hours. Fish were kept in 12˚C freshwater tanks for 23 days. Four fish per group were sampled at each time point (6 h.p.i, 1, 2, 3, 4, 5, 6, 7, 8, 10, 12, 14, 19, and 23 d.p.i, respectively). Specimen from moribund naturally ISAV-infected Atlantic salmon were obtained during clinical ISA outbreaks and were confirmed ISAV-positive by the Norwegian Veterinary Institute. To generate IHNV-infected tissues, 40 unvaccinated Atlantic salmon (mean weight 140g) were anesthetised with tricaine methanesulfonate (MS222, 80 mg/L) and exposed to IHNV by intraperitoneal injection of $10^6$ plaque forming units per fish, delivered in a single 100 μl dose. Fish were kept in 12˚C seawater (32ppt) and monitored daily for mortality and signs of disease for 10 days. Samples were obtained at the end of the trial, and viral RNA in kidneys was measured by qPCR as previously described [69], demonstrating heavy infection with >$10^5$ viral transcripts per μg RNA (S2 Table). Fish in both trials were euthanized with an overdose of MS222 (250 mg/L) before sampling.

## Virus binding assays

To map virus-binding sites in Atlantic salmon tissues, we used virus preparations or recombinant ISAV HE as probes as previously described [5] (Fig 1A). We would like to stress that the assay uses virus preparations and not antibodies as primary probes. Formalin-fixed paraffin-embedded tissues were heat treated (60–70˚C, 20 min), deparaffinised, incubated with 100 μL ISAV antigen (512 haemagglutinating units/mL, 60 min, RT) and washed in PBS. To detect virus binding, endogenous peroxide was quenched with peroxidase block (5 min, RT), sections were incubated with blocking buffer (5% dry milk or background sniper, 30 min, RT), then with mouse IgG$_1$ anti-ISAV HE (3H6F8 [70], 1/10-100, 60 min, RT), washed in PBS, and signal was visualised by the EnVision HRP-DAB rabbit/mouse (Agilent Technologies) or the MACH2 HRP polymer-DAB (Biocare Medical) systems, following manufacturer's instructions.

To measure the virus binding potential of cultured cells, ASK cells were cultured in 96-well cell culture plates and acetone-fixed (80%, 10 min, RT), incubated with 50 μL ISAV antigen

per well (512 haemagglutinating units/mL, 60 min, 4˚C), and washed three times with PBS. To detect bound viral antigen, cells were incubated with primary antibodies in 1xCMB (Thermo Fisher Scientific) (50 µL/well, 60 min, RT), washed three times in PBS, incubated with goat anti-mouse IgG-HRP (Thermo Fisher Scientific, 1:5000 in 1x CMB, 100 µL/well, 45 min, RT), washed 5 times in PBS, and incubated with TMB substrate (100 µL/well, 10 min, RT, in the dark). The reaction was stopped by adding $H_2SO_4$ (0.18M, 100 µL/well). The optical density (OD) at 450 nm was read by a Tecan sunrise plate reader and data extracted using Magellan software v7.2.

## Immunohistochemistry

Immunohistochemistry was performed as previously described [5], using rabbit antibody raised against ISAV NP (K716, 1:3000, 4˚C, overnight) or a mouse monoclonal antibody IgM targeting an epitope closely associated with the ISAV cellular binding site (10E4 [20], 1:100, RT, 60 min). Signal was visualised by the Vectastain ABC anti-rabbit IgG AP Immunodetection kit (Vector Laboratories), according to manufacturer's instructions. Detail about specific reagents is provided in S1 Table.

## Cellular ELISA

Infected ASK cells in 96-well plates were fixed by acetone (80%, 10 min, RT). The cellular expression of ISAV HE or IHNV was measured by ELISA: Cells were incubated with primary antibodies in 1xCMB (50 µL/well, 60 min, RT), washed three times in PBS, incubated with goat anti-mouse IgG-HRP (Thermo Fisher Scientific, 1:5000 in 1x CMB, 100 µL/well, 45 min, RT), washed 5 times in PBS, and incubated with TMB substrate (100 µL/well, 10 min, RT, in the dark). The reaction was stopped with $H_2SO_4$ (0.18M, 100 µL/well). The optical density (OD) at 450 nm was read by a Tecan sunrise plate reader and data extracted using Magellan software v7.2. Specific antibodies and dilutions are given in S1 Table.

## Immunofluorescent staining

For detection of viral antigens, ASK cells were fixed in acetone (80%, 10 min, RT) or paraformaldehyde (4%, 10 min, RT), washed, incubated with primary antibodies (60 min, RT), washed three times in PBS, incubated with goat anti-mouse IgG–Alexa 488 (Thermo Fisher Scientific, 45 min, RT), washed three times in PBS, counterstained with Hoechst 33342 (Sigma-Aldrich, 2 µg/mL, 3 min, RT), and stored and imaged in PBS or mounting medium (Ibidi GmbH). Specific antibodies and dilutions are given in S1 Table. For immunofluorescent staining of tissues, deparaffinized and heat treated (60–70˚C, 20 min) sections of formalin-fixed paraffin-embedded tissues were incubated with mouse monoclonal IgM targeting an epitope closely associated with the ISAV-binding epitope (10E4 [20], 1:100, RT, 60 min), washed in PBS, and incubated with goat anti-mouse IgM—Alexa594 (Thermo Fisher Scientific, 45 min, RT). Wide-field imaging was performed on a Zeiss Axio Observer A1 fluorescent microscope with a 40x LD-plan Neofluar 40x objective (N/A 0.6) or a Plan Neofluar 40x oil objective (N/A 1.3). Confocal imaging was performed on a Zeiss LSM710 confocal microscope with Plan Apochromat SF25 40x oil objective (N/A 1.3) or a Plan Apochromat 63x oil objective (N/A 1.4).

## Scoring of virus histochemistry and stainings

For scoring of virus receptor loss, sections subjected to virus histochemistry were blinded and scored from 0 to 5: (0) 100% viral receptor loss, (1) 80–99% viral receptor loss, (2) 60–80%

viral receptor loss, (3) 40–60% viral receptor loss, (4) 20–40% viral receptor loss, and (5) 0–20% viral receptor loss. Sections and blood smears subjected to ISAV were blinded and scored from 0 to 5 as previously described [39], but with 0 indicating absence of signal and 5 representing extensive detection of viral antigens. For scoring of 10E4 signal after sialidase treatment, sections subjected to 10E4 staining were blinded and scored from 0 to 4: (0) no signal, (1) reduced/speckled signal in occasional vessels, (2) reduced/speckled signal in many vessels, (3) medium strength, smooth vascular signal, (4) strong, smooth vascular signal.

### Statistics

Statistics for *in vitro* experiments were performed in Graph Pad Prism 8 for Windows 64-bits, v.8.4.3. Spearman correlation and bootstrapping were performed in R v.4.0.0, using R-package boot and basic bootstrap 95% confidence intervals.

## Supporting information

**S1 Fig. Supplemental to Fig 1.** Representative micrographs of virus histochemistry in sections from kidney (A) and liver (B) of fish infected with CA/F679/99 and NO/Glesvaer/2/90, respectively. (n = 1 fish per time group). Positive binding is identified by DAB (ISAV binding, light brown). The dark brown cellular signal in kidneys is pigment in melanomacrophages and should not be confused with positive signal.
(TIF)

**S2 Fig. Supplemental to Fig 2.** Representative micrographs of 10E4 immunofluorescent staining of sialic acid (red). (A) 10E4 staining of hearts of non-infected and NO/Glesvaer/2/90 infected (19 d.p.i.) fish showed no difference in signal intensity. (B) No 10E4 signal could be detected in tissue sections of non-infected fish pre-treated with NaOH and sialidase (negative control for Figs 2B and S2C). The extensive autofluorescence in kidney tubules should not be confused with positive signal. (C-D) 10E4 signal in kidney sections of experimentally infected and non-infected fish (n = 4 fish per group), confirming findings from heart, that NO/Glesvaer/2/90 infection renders the 10E4 epitope sensitive to sialidase.
(TIF)

**S3 Fig. Supplemental to Fig 3.** (A) NO/Glesvaer/2/90 and CA/NBISA05/98 co-infected ASK cells, (B) NO/Glesvaer/2/90 infected ASK cells, and (C) CA/NBISA05/98 infected ASK cells were fixed and immunostained with the hemagglutinin esterase (HE)-reactive antibody clones 10C9 (green) and 8F5 (red), specific to European and North-American ISAV genogroups, respectively.
(TIF)

**S1 Table. Reagents and antibodies.**
(DOCX)

**S2 Table. Numericals used for generating graphs and histograms.**
(XLSX)

## Acknowledgments

We would like to thank the engineers at the Section for Pathology and Section for Immunology and Virology at the Norwegian Veterinary Institute (Oslo, Norway) for technical assistance, the personnel at VESO Vikan and the aquaria staff at the Pacific Biological Station for

assistance with fish experiments, and Niels-Jørgen Olesen (EURL for Fish and Crustacean Diseases, DTU, DK) for providing IHNV antibodies.

## Author Contributions

**Conceptualization:** Maria Aamelfot, Johanna Hol Fosse, Knut Falk.

**Data curation:** Knut Falk.

**Formal analysis:** Maria Aamelfot, Johanna Hol Fosse, Knut Falk.

**Funding acquisition:** Johanna Hol Fosse, Alastair McBeath, Knut Falk.

**Investigation:** Maria Aamelfot, Johanna Hol Fosse, Frieda Betty Ploss, Sylvie L. Benestad, Debes Hammershaimb Christiansen, Kyle Garver, Knut Falk.

**Methodology:** Maria Aamelfot, Johanna Hol Fosse, Sylvie L. Benestad, Alastair McBeath, Knut Falk.

**Project administration:** Maria Aamelfot, Johanna Hol Fosse, Alastair McBeath, Debes Hammershaimb Christiansen, Knut Falk.

**Resources:** Maria Aamelfot, Alastair McBeath, Debes Hammershaimb Christiansen, Kyle Garver, Knut Falk.

**Supervision:** Knut Falk.

**Validation:** Maria Aamelfot, Johanna Hol Fosse.

**Visualization:** Maria Aamelfot, Johanna Hol Fosse, Hildegunn Viljugrein.

**Writing – original draft:** Maria Aamelfot, Johanna Hol Fosse, Knut Falk.

**Writing – review & editing:** Maria Aamelfot, Johanna Hol Fosse, Hildegunn Viljugrein, Frieda Betty Ploss, Sylvie L. Benestad, Alastair McBeath, Debes Hammershaimb Christiansen, Kyle Garver, Knut Falk.

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
