## [Decision Letter · Decision Letter 0]

22 Aug 2022

Dear Dr. Hol,

Thank you very much for submitting your manuscript "Destruction of the vascular viral receptor in infectious salmon anaemia provides in vivo evidence of homologous attachment interference" for consideration at PLOS Pathogens. As with all papers reviewed by the journal, your manuscript was reviewed by members of the editorial board and by several independent reviewers. The reviewers appreciated the attention to an important topic. Based on the reviews, we are likely to accept this manuscript for publication, providing that you modify the manuscript according to the review recommendations.

All three reviewers were positive about the paper but made some constructive suggestions for improvement. Please consider these comments in turn.

Sincerely,

Anice C. Lowen

Associate Editor

PLOS Pathogens

Ron Fouchier

Section Editor

PLOS Pathogens

Kasturi Haldar

Editor-in-Chief

PLOS Pathogens

orcid.org/0000-0001-5065-158X

Michael Malim

Editor-in-Chief

PLOS Pathogens

orcid.org/0000-0002-7699-2064

All three reviewers were positive about the paper but made some constructive suggestions for improvement. Please consider these comments in turn.

Reviewer Comments (if any, and for reference):

Reviewer's Responses to Questions

**Part I - Summary**

Reviewer #1: The manuscript by Maria Aamelfot describes that ISAV infection removes viral receptors that as such the fish become refractory to a subsequent infection. ISAV is a member of the orthomyxoviridea carrying an HE and fusion envelope protein. The HE protein is special as it binds and cleaves 4-0-Ac sialic acids. I my knowledge only MHV HE has a similar specificity and activity (not referenced). Especially nice is the demonstration that a heterologous virus infection does not inhibit ISAV infection.

The manuscript is of great interest and I for sure would recommend it for publication in Plos Path and only have several suggestions to improve the paper.

Although the figures are presented nicely, they could be improved to follow them without text. Especially from fig 4 and the supplemental in which it is hard to follow what is stained (which tissue / which antibody specific for x?) and what the treatment was, for most readers VHC, RBC IFAT, and VBA block become a puzzle. I’m also not such a fan of the different scoring systems, why is this not quantifiable? Panel E is interesting but counterintuitive as 1% means an increase and the other a decrease. Figure 4 could thus use an overhaul.

The authors nicely reference old-school Schauer literature, but recent literature is dotted with gems describing specific 0-acetyl binding by HE and spike.

Imho the introduction and discussion should be condensed to make it easier to follow and a quicker read. I understand the fact that a homologous infection prevents a secondary by removing a particular sialic acid is of interest, but it is a bit of an open door as well. The discussion goes from one topic to the other and back. I would like to read something more focused on why this 4-0Ac sialic acid is of such interest as viral receptors. For example horse a-macroglobulin is extremely rich in 4-OAc and historically used as an inhibitor (with very high concentrations) for influenza A viruses, would this now be a perfect inhibitor for ISAV?

I know this is hard but are there no specific lectins that only bind 4-0Ac? The 10e4 antibody does not depend on the 4-0Acetyl, and fig 2B top middle panel gives a distinct binding pattern? An ISAV HE-fc with enzymatic knockout would be a great way to go (or MHV?).

Robert de Vries

Utrecht Institute for Pharmaceutical Sciences

Reviewer #2: This study by Aamelfot, Hol Fosse and colleagues examines the broadly important question of viral superinfection exclusion (ie, how a virally infected cell is able to prevent subsequent infection by a homologous virus) using infectious salmon anemia virus (ISAV). The authors show that this resistance to infection follows the de-acetylation of sialic acids on the cell surface, in the relevant animal model during the full course of infection/disease. The destruction of the viral surface receptor was able to prevent re-infection with a related ISAV strain, and the authors demonstrated that this interference was specific to ISAV (as prior infection with either a different RNA virus, IHNV, or pre-treatment with Poly I:C was not able to prevent ISAV infection). The question of how and why virally-infected cells are able to prevent superinfection by homologous virions has important implications for the field and this study provides a valuable look at how this process occurs for a segmented RNA virus in the in vivo host. It would be very interesting to see the direct effect of transfecting HE (or more specifically the esterase portion) on the stability of the viral receptor, and whether the downregulation observed in infection occurs in the context of this protein alone. The authors elegantly look at receptor degradation in the context of days-to-weeks, over the entire course of infection. It would be very interesting to look at the kinetics of receptor loss early after infection as well, to determine the first time at which degradation/superinfection exclusion occurs (and whether the two happen at the same time over the very early times of infection or if there is a separation in relative kinetics that suggests multiple mechanisms of SIE).

Reviewer #3: PPATHOGENS-D-22-01074

Destruction of the vascular viral receptor in infectious salmon anaemia provides in vivo

evidence of homologous attachment interference

In this ms the authors have investigated the possible role of ISAV hemagglutinin esterase (HE) as a mediator of viral interference in Atlantic salmon infected with ISAV. Viral interference (one infection inhibits another) is an understudied phenomenon in virology and certainly regarding fish viruses. The authors have developed a virus histochemistry method facilitating the quantification of tissue attachment sites for ISAV by measuring binding of virus to tissue sections. Using this and complementary methods the authors test the hypothesis that when ISAV infects endothelial cells of Atlantic salmon, it restricts subsequent homologous infection by removing the 4-O-acetyl group necessary for binding from sialic acid. Interesting differences in this activity between high and low virulent isolates of ISAV is measured and discussed in relation to pathogenesis. The loss of viral receptors upon infection reduce coinfection with other strains and limit the extent of host damage.

This study is well planned and conducted with relevant controls and sufficient replications. The results are presented in clear figures with appropriate statistics.

The presented data supports the authors claims

It is certainly of interest to virologists in general and of special interest to fish disease scientists.

**Part II – Major Issues: Key Experiments Required for Acceptance**

Reviewer #1: none

Reviewer #2: I think the conclusions of the study are well supported by the data. I think the paper would be strengthened if the authors included slightly more data tying the phenomenon directly to HE protein expression, and/or mapping the induction of SIE vs receptor loss at a finer level of kinetic resolution early in infection in cultured cells.

Reviewer #3: None

**Part III – Minor Issues: Editorial and Data Presentation Modifications**

Reviewer #1: See Part I

Reviewer #2: It would help readability if the authors included the identity of each antibody used in confocal images within the figure panel rather than only in the legend.

Reviewer #3: Minor points:

Figure 1B-C - visualizing a decreasing parameter (binding) as an increase in "Loss of binding" is not optimal. Why not plot "Residual binding" as a falling curve?. Its more logical. And why not plot the average residual binding at each timepoint a points on a curve (as in figure 4)?

Figure 3. Infection of ASK cells. There are esterase inhibitors available that coud be tested for activity and effect in vitro. Any particular reason this was not included?

Fig S2 D Align subscripts (skewed to the right on the screen version but looks better when printed)

PLOS authors have the option to publish the peer review history of their article (what does this mean?). If published, this will include your full peer review and any attached files.

Reviewer #1: No

Reviewer #2: No

Reviewer #3: No

Figure Files:

Data Requirements:

Reproducibility:

References:

---

## [Editor Report · Decision Letter 1]

1 Oct 2022

Dear Dr. Hol,

We are pleased to inform you that your manuscript 'Destruction of the vascular viral receptor in infectious salmon anaemia provides in vivo evidence of homologous attachment interference' has been provisionally accepted for publication in PLOS Pathogens.

Best regards,

Anice C. Lowen

Associate Editor

PLOS Pathogens

Ron Fouchier

Section Editor

PLOS Pathogens

Kasturi Haldar

Editor-in-Chief

PLOS Pathogens

orcid.org/0000-0001-5065-158X

Michael Malim

Editor-in-Chief

PLOS Pathogens

orcid.org/0000-0002-7699-2064

Thank you for your thorough revisions addressing the reviewers' comments.
---

## [Editor Report · Acceptance letter]

11 Oct 2022

Dear Dr Falk,

We are delighted to inform you that your manuscript, "Destruction of the vascular viral receptor in infectious salmon anaemia provides in vivo evidence of homologous attachment interference," has been formally accepted for publication in PLOS Pathogens.

Best regards,

Kasturi Haldar

Editor-in-Chief

PLOS Pathogens

orcid.org/0000-0001-5065-158X

Michael Malim

Editor-in-Chief

PLOS Pathogens

orcid.org/0000-0002-7699-2064